# Scorpionfish BPI is highly active against multiple drug-resistant *Pseudomonas aeruginosa* isolates from people with cystic fibrosis

Jonas Maurice Holzinger[1], Martina Toelge[1], Maren Werner[1], Katharina Ursula Ederer[1], Heiko Ingo Siegmund[2], David Peterhoff[1,3], Stefan Helmut Blaas[4], Nicolas Gisch[5], Christoph Brochhausen[2,6], André Gessner[1,3], Sigrid Bülow[1]*

[1]Institute of Clinical Microbiology and Hygiene Regensburg, University Hospital Regensburg, Regensburg, Germany; [2]Institute of Pathology, University of Regensburg, Regensburg, Germany; [3]Institute of Medical Microbiology and Hygiene Regensburg, University of Regensburg, Regensburg, Germany; [4]Department of Pneumology, Donaustauf Hospital, Donaustauf, Germany; [5]Division of Bioanalytical Chemistry, Priority Area Infections, Research Center Borstel, Leibniz Lung Center, Borstel, Germany; [6]Institute of Pathology, University Medical Center Mannheim, Medical Faculty Mannheim, University of Heidelberg, Mannheim, Germany

**\*For correspondence:**
sigrid.buelow@ukr.de

**Abstract** Chronic pulmonary infection is a hallmark of cystic fibrosis (CF) and requires continuous antibiotic treatment. In this context, *Pseudomonas aeruginosa* (*Pa*) is of special concern since colonizing strains frequently acquire multiple drug resistance (MDR). Bactericidal/permeability-increasing protein (BPI) is a neutrophil-derived, endogenous protein with high bactericidal potency against Gram-negative bacteria. However, a significant range of people with CF (PwCF) produce anti-neutrophil cytoplasmic antibodies against BPI (BPI-ANCA), thereby neutralizing its bactericidal function. In accordance with literature, we describe that 51.0% of a total of 39 PwCF expressed BPI-ANCA. Importantly, an orthologous protein to human BPI (huBPI) derived from the scorpionfish *Sebastes schlegelii* (scoBPI) completely escaped recognition by these autoantibodies. Moreover, scoBPI exhibited high anti-inflammatory potency towards *Pa* LPS and was bactericidal against MDR *Pa* derived from PwCF at nanomolar concentrations. In conclusion, our results highlight the potential of highly active orthologous proteins of huBPI in treatment of MDR *Pa* infections, especially in the presence of BPI-ANCA.

## Editor's evaluation

In this useful study, Holzinger et al. present compelling evidence that scorpionfish bactericidal/permeability-increasing protein (scoBPI) exhibits remarkable antibacterial activity against multi-drug resistant *Pseudomonas aeruginosa*. These findings open new avenues of research for identifying novel chemotherapies to treat *Pseudomonas* infections and have broader implications in developing chemotherapies against other drug-resistant Gram-negative bacterial infections. The work will be of interest to individuals investigating novel cystic fibrosis antimicrobials.

**eLife digest** Cystic fibrosis is a genetic disorder that makes people produce unusually thick and sticky mucus that clogs their lungs and airways. This inevitably leads to recurring bacterial infections, particularly those caused by the Gram-negative bacterium *Pseudomonas aeruginosa*. Antibiotics are needed to treat these infections. However, over time most bacteria build modes of resistance to these drugs and, once multiple drug-resistant bacteria colonize the lung, very limited treatment options are left. Therefore, new therapeutic approaches are desperately needed.

Notably, humans themselves express a highly potent antimicrobial protein called BPI (short for Bactericidal/permeability-increasing protein) that attacks Gram-negative bacteria, including multiple drug-resistant strains of *P. aeruginosa*. Unfortunately, many people with cystic fibrosis also generate antibodies that bind to BPI and interfere with its antimicrobial function.

Faced with this conundrum, Holzinger et al. set out to find BPIs made by other animals which might not be recognized by human antibodies and also display a high potential to attack Gram-negative bacteria. Based on specific selection criteria, Holzinger et al. focused their attention on BPI made by scorpionfish, a type of venomous fish that live near coral reefs.

Compared to other BPI proteins they investigated, the one produced by scorpionfish appeared to be the most capable of binding to *P. aeruginosa* via a prominent surface molecule exclusively found on Gram-negative bacteria. Furthermore, when Holzinger et al. tested whether the antibodies present in people with cystic fibrosis could recognize scorpionfish BPI, they found that the BPI completely evaded detection. The scorpionfish BPI was also able to pre-eminently attack *P. aeruginosa*. In fact, it was even able to potently kill drug-resistant strains of the bacteria that had been isolated from people with cystic fibrosis.

This study suggests that scorpionfish BPI could serve as an alternative to antibiotics in people with cystic fibrosis that have otherwise untreatable bacterial infections. Drug-resistant bacteria which cause life threatening conditions are on the rise across the globe, and scorpionfish BPI could be a potential candidate to treat affected patients. In the future, animal experiments will be needed to explore how highly potent non-human BPIs function in whole living organisms.

## Introduction

Cystic fibrosis (CF) is an autosomal-recessive disorder induced by loss-of-function mutations in the gene encoding for cystic fibrosis transmembrane conductance regulator (CFTR) protein (*Collins, 1992*). Defects in *CFTR* negatively influence homeostasis of airway surface liquid and mucus of the respiratory epithelium, aggravating mucociliary clearance (*Boucher, 2007*; *Gustafsson et al., 2012*). Despite the clinical improvements achieved by CFTR modulators in people with at least one F508del mutation (*Middleton et al., 2019*), chronic pulmonary infections with the opportunistic pathogen *Pseudomonas aeruginosa* (*Pa*) will continue to determine disease progression in untreated or unresponsive CF (*Koch, 2002*; *Emerson et al., 2002*). In these cases, continuous suppressive antibiotic therapy is a standard regime to improve survival. However, the long-term application of antibiotics promotes the emergence of persister populations (*Balaban et al., 2019*; *Rossi et al., 2021*) and multiple drug-resistant (MDR) bacteria (*Pitt et al., 2003*), thus limiting the therapeutic options. The neutrophil-derived bactericidal/permeability-increasing protein (BPI) is highly active against Gram-negative bacteria (*Weiss et al., 1978*; *Weiss et al., 1992*). Additionally, BPI acts as an anti-inflammatory mediator by neutralizing Gram-negative lipopolysaccharide (LPS; *Dentener et al., 1993*; *Weiss et al., 1983*). By inhibiting growth of *Escherichia coli* (*Ec*) and neutralizing endotoxic activity of LPS in a picomolar range (*Ederer et al., 2022*), it is one of the most effective antimicrobials in the human body. The recombinant N-terminal fragment of human BPI (rBPI21) was shown to be highly effective against MDR Gram-negative bacteria, including MDR *Pa* (*Weitz et al., 2013*). However, anti-neutrophil cytoplasmic autoantibodies directed against BPI (BPI-ANCA) found in up to 83% of people with CF (PwCF; *Theprungsirikul et al., 2021b*) inhibit the bactericidal function of BPI (*Schultz et al., 2004*; *McQuillan et al., 2020*). Coherently, BPI-ANCA correlate with increased colonization with *Pa* (*Hovold et al., 2020*), impaired lung function and poor prognosis (*Carlsson et al., 2007*). To restore BPI function, we speculated that orthologous proteins with conserved bactericidal and LPS-neutralizing activity would escape recognition by intrinsic BPI-ANCA and therefore preserve their bactericidal activity when

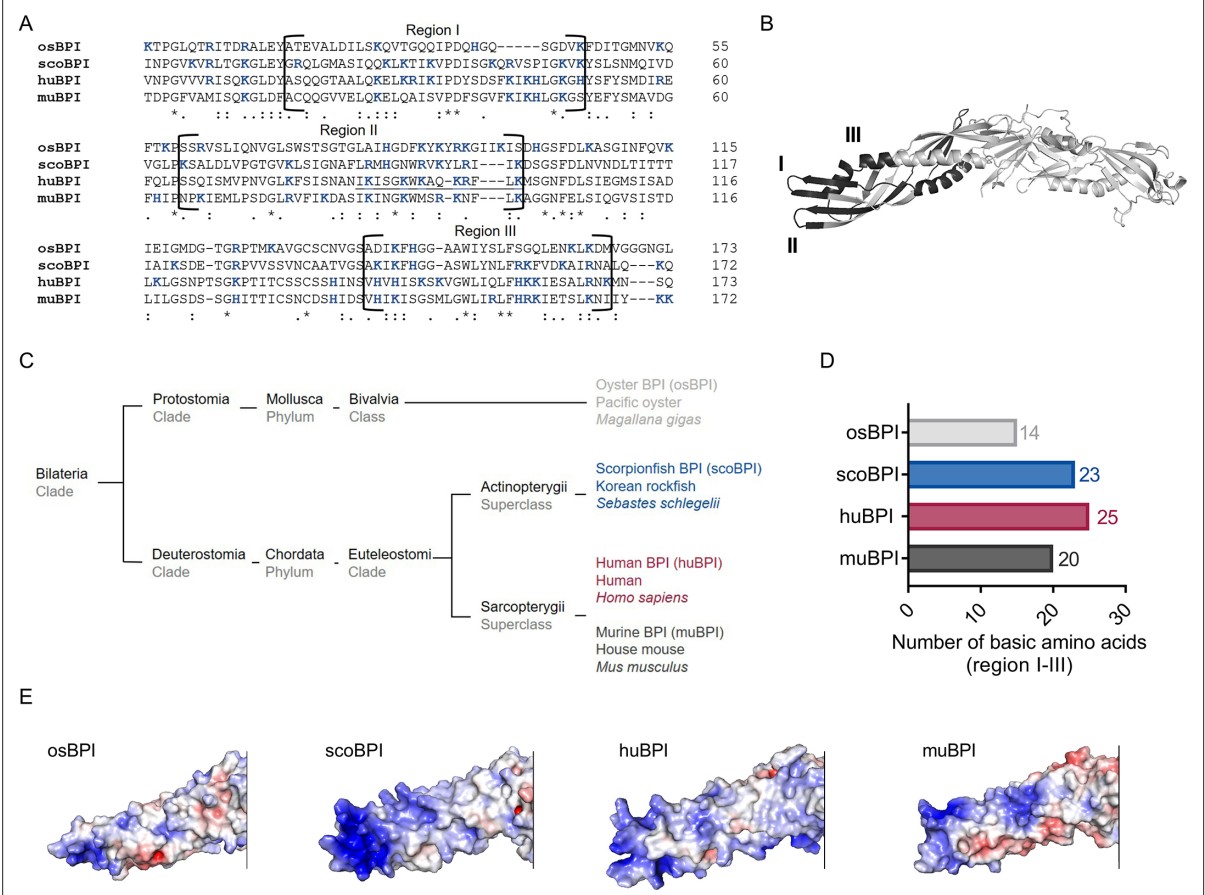

**Figure 1.** Sequence alignment and analysis of orthologous proteins to human bactericidal/permeability-increasing protein (huBPI). (**A**) Sequence alignment of amino acid sequences of orthologous proteins. Functional regions I–III are framed, and positively charged amino acids are shown in blue. The amino acid sequence mediating the bactericidal activity of human BPI (huBPI) is underlined. (**B**) Functional regions I–III (dark grey) as shown for huBPI. (**C**) Phylogenetic tree analysis mapping BPI orthologous proteins to their respective taxonomic ranks. (**D**) Number of basic amino acids in regions I–III as shown for oyster BPI (osBPI), scorpionfish BPI (scoBPI), huBPI and murine BPI (muBPI). (**E**) Electrostatic surface potentials calculated for the N-terminal barrel of osBPI, scoBPI, huBPI, and muBPI. Negatively charged areas are colored in red, and positively charged domains are shown in blue.

The online version of this article includes the following source data and figure supplement(s) for figure 1:

**Figure supplement 1.** Sequence alignment and analysis of orthologous Actinopterygii bactericidal/permeability-increasing protein (BPI).

**Figure supplement 2.** Predicted N-terminal surface electrostatics and sequence identities of Actinopterygii bactericidal/permeability-increasing protein (BPI).

**Figure supplement 2—source data 1.** Uncropped and labeled images for *Figure 2—figure supplement 1C*.

applied to BPI-ANCA-positive PwCF. Based on known properties, sequence homologies, and structural considerations, we selected murine BPI (muBPI) of the house mouse *Mus musculus* (**Wittmann et al., 2008**), scorpionfish BPI (scoBPI) of the Korean rockfish *Sebastes schlegelii* (**Lee et al., 2017**), and oyster BPI (osBPI) of *Magallana gigas* (Pacific oyster, previously *Crassostrea gigas*; **Gonzalez et al., 2007**; **Zhang et al., 2011**) for comparison to human BPI (huBPI, *Homo sapiens*). In contrast to huBPI, neither muBPI nor scoBPI were recognized by BPI-ANCA present in the sera of PwCF. Furthermore, scoBPI displayed superior LPS-neutralizing capacity and bactericidal activity towards MDR *Pa* at a nanomolar range.

## Results

### Comparison of the N-terminal barrel in orthologous BPI

The N-terminal domain of huBPI contains three highly cationic regions (regions I–III; *Figure 1A and B*), which engage with the negatively charged moieties of LPS to neutralize its pro-inflammatory potential

(*Little et al., 1994*; *Gazzano-Santoro et al., 1995*). Therefore, we hypothesized that the affinity towards LPS is roughly determined by the respective quantity of basic amino acids. Preserved LPS neutralization as well as bactericidal activity was previously described for BPI orthologues derived from distantly related species such as the Pacific oyster (*Gonzalez et al., 2007*; *Zhang et al., 2011*) and ray finned fish (Actinopterygii; *Lee et al., 2017*; *Sun and Sun, 2016*). In comparison to nine other Actinopterygii BPI sequences, scoBPI comprised the highest number of the cationic amino acids arginine, histidine, and lysine in the corresponding regions (*Figure 1—figure supplement 1A–C*). A sequence alignment also revealed that scoBPI contained more basic amino acids in regions I–III than muBPI and osBPI (*Figure 1A, B and D*). In contrast to LPS binding, only 15 amino acids in region II determine the antimicrobial activity in huBPI (*Little et al., 1994*; *Figure 1A*). Within this region, sequence analysis revealed a conserved cationic charge with a total of five basic amino acids for muBPI or six for osBPI, scoBPI, and huBPI, respectively (*Figure 1A*). Exploiting the published three-dimensional structures and electrostatic surface potentials for huBPI (*Beamer et al., 1997*) and AlphaFold Protein Structure Database (*Jumper et al., 2021*; *Varadi et al., 2022*), predictions for the orthologous proteins displayed a highly condensed positively charged cluster in scoBPI that was exclusively found at the N-terminal tip (*Figure 1E*). Within the investigated class of Actinopterygii, acBPI of the sterlet *Acipenser ruthenus* (subclass Chondrostei) and gaBPI of the Atlantic cod *Gadus morhua* (subclass Neopterygii) also presented a cluster of positive charges in the respective area along with a relatively high quantity of basic amino acids in functional regions I–III (*Figure 1—figure supplements 1B and 2A*). Interestingly, overall sequence identity with scoBPI was comparably low (56.4 and 62.8%, respectively; *Figure 1—figure supplement 2B*). In conclusion, this initial screen revealed a higher quantity of positively charged surface areas for scoBPI and other fish BPI compared to the mammalian muBPI or the distantly related osBPI.

## No recognition of scoBPI by BPI-ANCA from PwCF

As stated, the recognition of endogenous BPI by BPI-ANCA is known to inhibit its bactericidal function and predict disease progression in PwCF (*Carlsson et al., 2007*). Sequence homology between huBPI and Actinopterygii orthologues, including scoBPI, was low ranging from 36.2 to 38.9% (*Figure 2A*, *Figure 1—figure supplement 2B*), and, consistent with distant or close relationship to huBPI, 24.9% for osBPI and 54.3% for muBPI, respectively (*Figure 2A*). Therefore, we speculated that BPI-ANCA in the sera of PwCF would not bind to non-huBPI. First, we screened the sera of 39 PwCF for the presence of BPI and BPI-ANCA. While levels of BPI in the sera were comparable among PwCF and an age- and sex-matched control group (*Figure 2B*), BPI-ANCA as measured in arbitrary units (AU) were significantly increased in PwCF (control: mean 314.8 AU, 95% confidence interval [CI] 209.3–420.4 AU; CF: mean 2218.0 AU, 95% CI 1271.0–3166.0 AU, p<0.0001; *Figure 2C*). Based on our evaluation described above, we selected muBPI, scoBPI, and osBPI for functional analysis. After recombinant expression of huBPI, muBPI, scoBPI, and osBPI (*Figure 2D*, *Figure 2—figure supplement 1A–C*), we used beads coupled with the respective BPI orthologues to analyze BPI-ANCA interaction (*Figure 2—figure supplement 1D*). While huBPI was bound by anti-BPI antibodies in 51.0%, muBPI and scoBPI were not recognized at all and osBPI was only detected by 2.6% of the CF sera (*Figure 2E*). Thus, the orthologous proteins muBPI and scoBPI do not contain epitopes detected by intrinsic BPI-ANCA of the tested cohort of PwCF.

## Superior anti-inflammatory activity of scoBPI in human peripheral blood mononuclear cells

Persistent bacterial colonization of PwCF, including Gram-negative bacteria and particularly *Pa*, drives chronic pulmonary inflammation and tissue damage (*Rossi et al., 2021*). The main immunostimulatory component of Gram-negative bacteria is LPS. Importantly, *Ec*-derived LPS can effectively be neutralized by picomolar concentrations of huBPI (*Ederer et al., 2022*). Therefore, we wanted to investigate whether the recombinant BPI orthologues were competent to interact with *Ec* LPS and neutralize its immunostimulatory potential. Importantly, we found that secretion of interleukin (IL)-6 in human peripheral blood mononuclear cells (PBMCs) by *Ec* LPS was most effectively inhibited by scoBPI, exceeding huBPI, muBPI, and osBPI (*Figure 3A*). This superior LPS-neutralizing capacity compared to huBPI could be observed for low picomolar concentrations of scoBPI (*Figure 3B*). To explore interacdtion of huBPI or orthologues with LPS derived from *Pa*, we purified LPS of *Pa* PAO1. Similar to

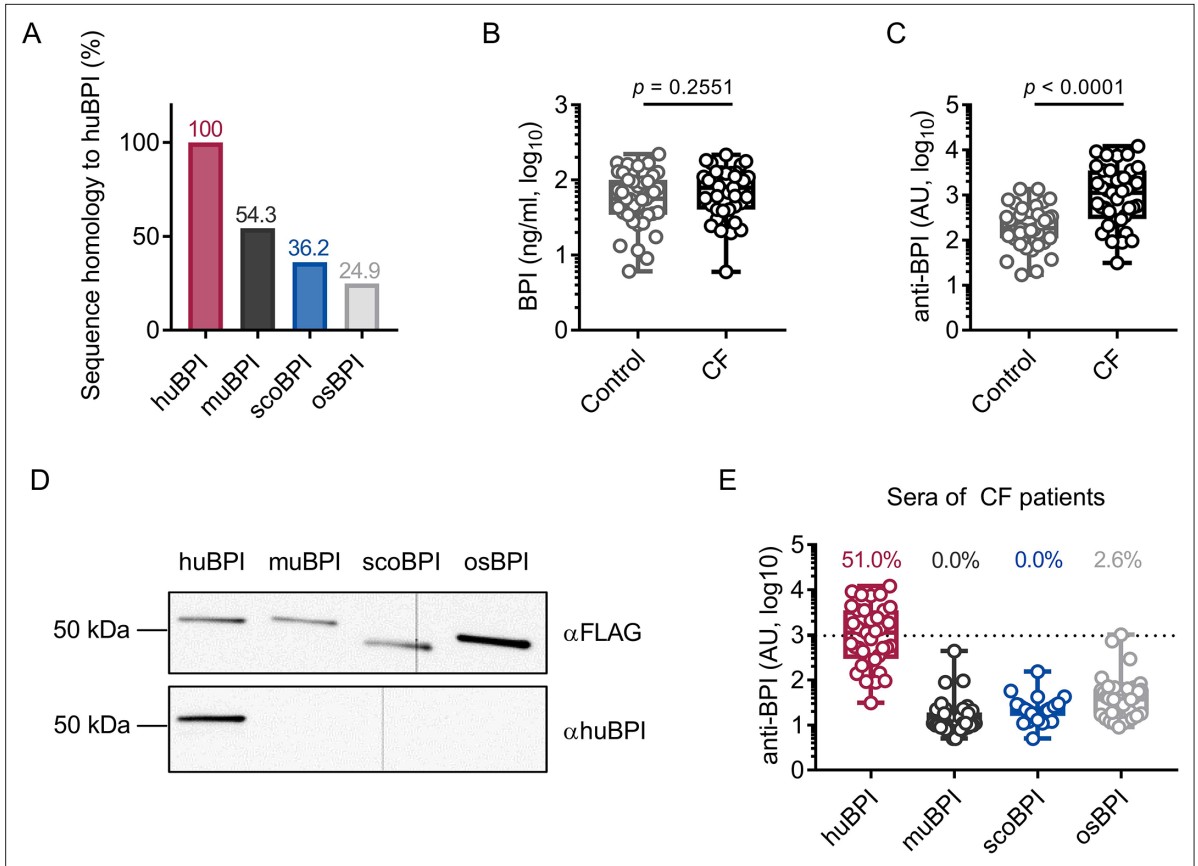

**Figure 2.** Bactericidal/permeability-increasing protein-anti-neutrophil cytoplasmic antibodies (BPI-ANCA) from people with cystic fibrosis (PwCF) do not recognize orthologous proteins of human BPI (huBPI). (**A**) Sequence homology of amino acid sequences to huBPI without signal peptide. (**B, C**) Levels of BPI (**B**) and anti-BPI antibodies indicated in arbitrary units (AU; **C**) in the sera of PwCF and age- and sex-matched healthy controls (n = 39). (**D**) Western blot analysis of recombinantly expressed proteins. Bands were visualized with anti-FLAG (αFLAG) or anti-human BPI (αhuBPI) antibodies. One representative experiment of two biological replicates using different protein lots in separate assays is shown. (**E**) Recognition of recombinantly expressed proteins by anti-BPI antibodies present in the sera of the individual PwCF shown in (**C**, n = 39). The signal cutoff is indicated by a dotted line and was defined as 2 standard deviations above the mean signal determined in the sera of healthy controls shown in (**C**). Data are shown as box plots showing median, upper, and lower quartiles and whiskers indicating minimal and maximal values (**B, C, E**) or bar plots using calculated values (**A**). Sera measurements were performed with samples from 39 individual PwCF and 39 age- and sex-matched controls. For depiction in the logarithmic scale, values of zero were set to the lower limit of detection of the assay (**B**). Statistical testing was performed using the Mann–Whitney *U* test. Statistical significance is indicated by p-values.

The online version of this article includes the following source data and figure supplement(s) for figure 2:

**Source data 1.** Raw data for *Figure 2B, C and E*.

**Figure supplement 1.** Purification of bactericidal/permeability-increasing protein (BPI) orthologues and quality controls.

**Figure supplement 1—source data 1.** Uncropped and labeled images for *Figure 2—figure supplement 1C*.

---

incubation of huBPI and LPS derived from *Salmonella minnesota* Re595 (*Tobias et al., 1997*), incubation of both huBPI and scoBPI with *Pa* LPS fostered prominent increase in aggregate size as measured by dynamic light scattering (DLS; *Figure 3C*) indicating close interaction. Congruent with aggregate formation, the pro-inflammatory potential of *Pa* LPS was drastically inhibited by huBPI and scoBPI (*Figure 3D and E*). In comparison to huBPI, scoBPI exhibited a superior anti-inflammatory potential as indicated by diminished production of IL-6 and tumor necrosis factor (TNF) in response to *Pa* LPS. These data suggest that scoBPI holds strong anti-inflammatory properties towards *Ec* and *Pa* LPS.

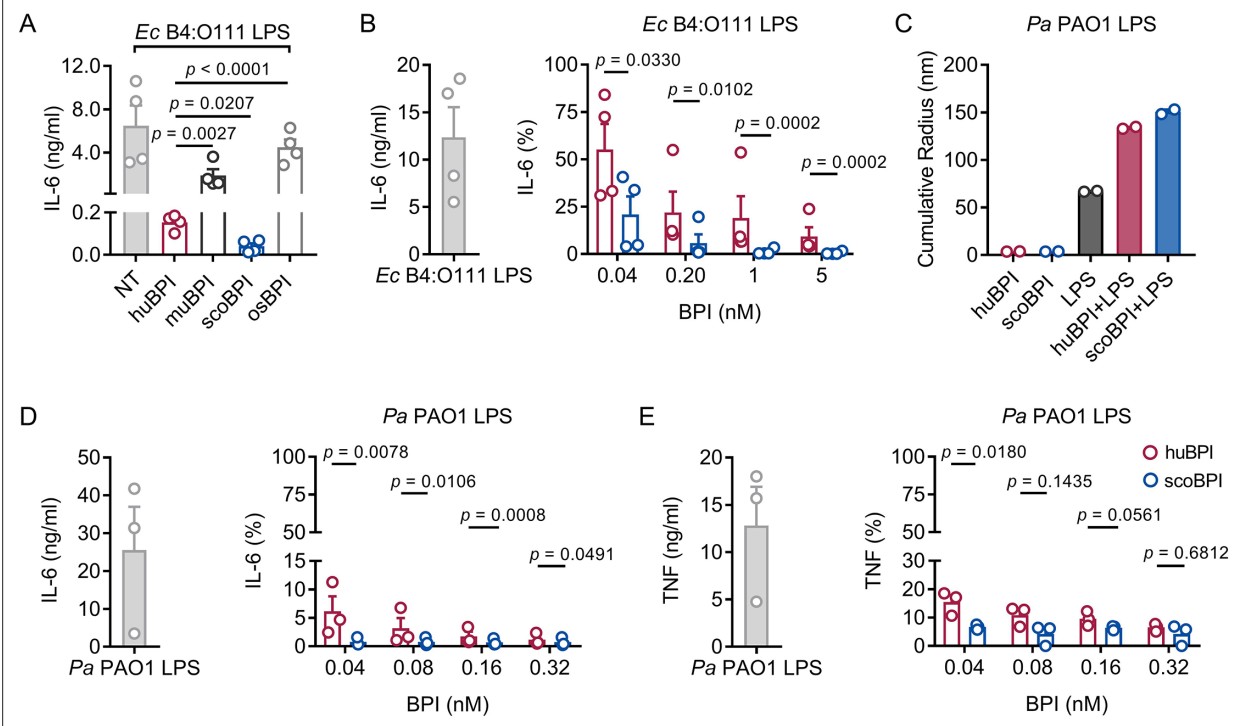

**Figure 3.** Potent anti-inflammatory action of scorpionfish bactericidal/permeability-increasing protein (scoBPI) in human immune cells. (**A**) Levels of IL-6 in supernatants of peripheral blood mononuclear cells (PBMCs) stimulated for 24 hr with *Ec* B4:O111 lipopolysaccharide (LPS; 10 ng/ml) ± human BPI (huBPI), murine BPI (muBPI), scoBPI, or oyster BPI (osBPI; 25 nM; n = 4). (**B**) Levels of IL-6 in supernatants of PBMCs stimulated for 24 hr with *Ec* B4:O111 LPS (10 ng/ml) ± huBPI (red) or scoBPI (blue) in concentration as indicated. (**C**) Aggregate size of *Pa* PAO1 LPS ± huBPI or scoBPI as determined by NanoDLS (n = 2). (**D, E**) Quantification of IL-6 (**D**) and TNF (**E**) levels in supernatants of PBMCs stimulated for 24 hr with *Pa* PAO1 LPS (100 ng/ml)±huBPI (red) or scoBPI (blue) in concentrations as indicated (n = 3). Experiments were performed using PBMCs of four (**A, B**) or three (**D, E**) individual blood donors. NanoDLS experiments were performed as technical replicates in two separate assays (**C**). Data are shown as means ± SEM. Statistical testing was performed using the Student's ratio paired *t*-test (**A, B, D, E**). Statistical significance is indicated by p-values.

The online version of this article includes the following source data for figure 3:

**Source data 1.** Raw data for *Figure 3A–E*.

## Bactericidal activity of scoBPI against multiple drug-resistant isolates of *Pseudomonas aeruginosa*

To evaluate the antimicrobial activity of the recombinantly expressed orthologous BPI proteins, we first compared their capacity to inhibit growth of *Ec* DH10B. Activity of huBPI (IC$_{50}$ 0.032 ± 0.011 nM), scoBPI (IC$_{50}$ 0.032 ± 0.006 nM), and muBPI (IC$_{50}$ 0.061 ± 0.016 nM) was in low picomolar range, while osBPI (IC$_{50}$ 60.16 ± 21.43 nM) required higher nanomolar concentrations to fully inhibit bacterial growth (*Figure 4A*). Bactericidal assays with *Pa* PAO1 revealed impaired activity of muBPI and osBPI, while huBPI and scoBPI potently inhibited bacterial growth (*Figure 4B*). Next, we were interested in whether BPI was also active against the highly adapted *Pa* strains from PwCF. Therefore, six distinct MDR *Pa* isolates of five PwCF were selected (mean age at sample collection 24.8 ± 7.1 y). Broad-spectrum resistance, including resistance to Ciprofloxacin, Piperacillin, Piperacillin/Tazobactam, Ceftazidime, Cefepime, Imipenem, and Meropenem, was verified by disc diffusion testing (*Table 1*). Despite major adaptions during chronic *Pa* infections (*Rossi et al., 2021*), three of the six isolates were also highly susceptible to both huBPI and scoBPI at concentrations of 20 nM after 1 hr of incubation (*Figure 4C*, *Figure 4—figure supplement 1A and B*). More than 50% growth reduction was achieved in all isolates within 1 hr at a concentration of 500 nM of scoBPI. Thereby, a trend towards superior antibiotic activity against MDR *Pa* for scoBPI compared to huBPI was observed (p = 0.0898). One of the less sensitive strains (isolate 5) potentially emerged from a previous isolate (isolate 2) in a time course of 3.3 y, possibly indicating adaptions to a long-term BPI exposure in vivo. In accordance with literature for huBPI (*Aichele et al., 2006*), mucoid strains were susceptible to both huBPI and scoBPI.

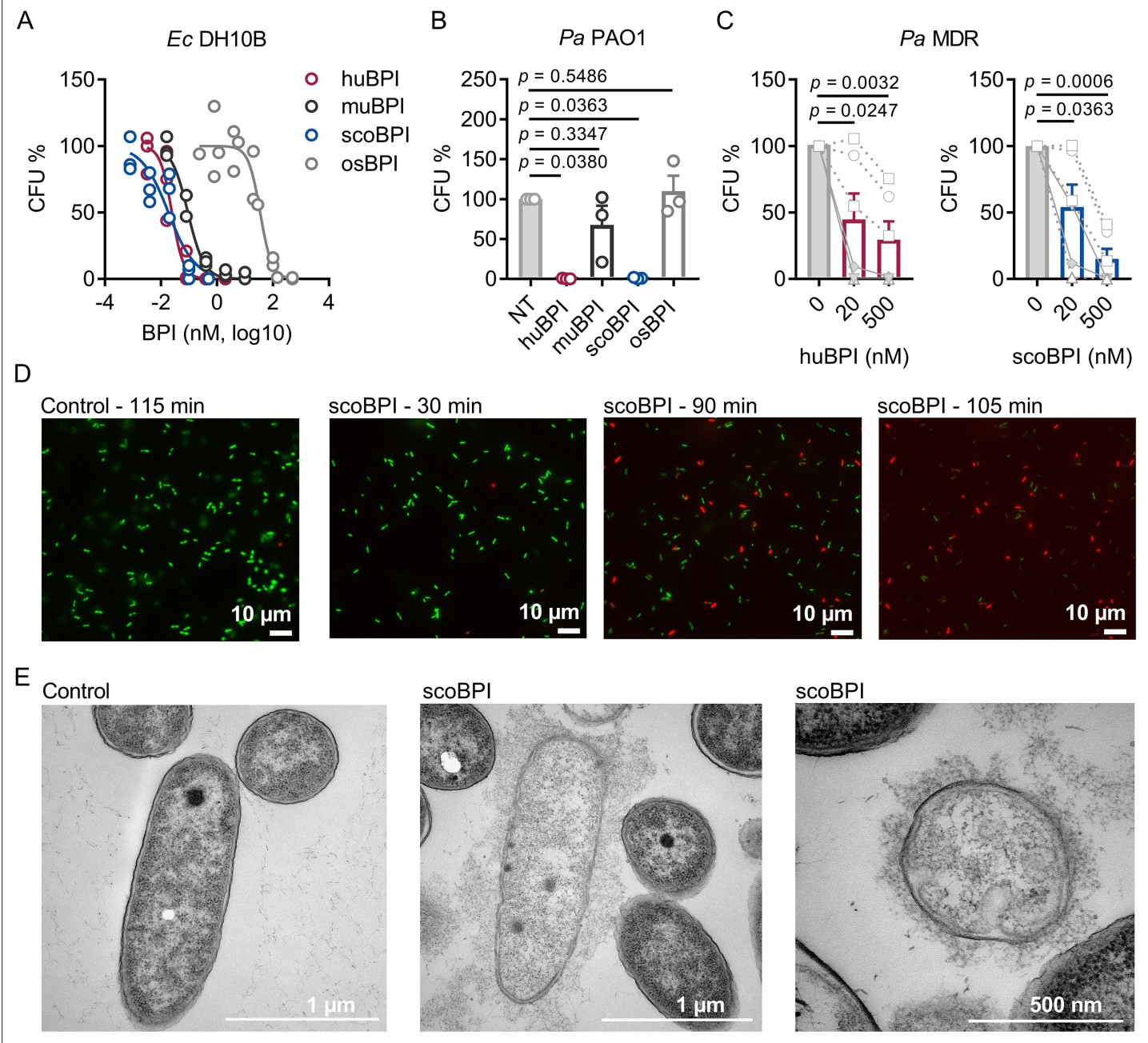

**Figure 4.** Bactericidal activity of human bactericidal/permeability-increasing protein (huBPI) and scorpionfish BPI (scoBPI) against *Ec* and multiple drug resistance (MDR) isolates of *Pa*. (**A**) Dose–response curves for *Ec* DH10B incubated with increasing concentrations of huBPI, murine BPI (muBPI), scoBPI, or oyster BPI (osBPI; n = 3). (**B**) Bactericidal activity of huBPI, muBPI, scoBPI, and osBPI 500 nM against *Pa* PAO1 (n = 3). (**C**) Antibacterial activity of huBPI and scoBPI at concentrations of 20 and 500 nM against six MDR isolates of *Pa* obtained from five individual people with cystic fibrosis (PwCF). Isolates (isolates 2 and 5 in **Table 1**) that originate from the same donor are shown as a square. Mucoid isolates are displayed as filled symbols connected by continuous lines. (**D**) Images of a bacterial viability assay performed with one representative *Pa* MDR strain. Viable bacteria are seen in green and dead bacteria in red, respectively. Scale bars for 10 μm. (**E**) Transmission electron microscopy of the representative *Pa* MDR isolate shown in (**D**) after treatment with PBS (control) or scoBPI for 2 hr. Scale bars for 1 μm (left, middle) or 500 nm (right). Data are shown as individual values (**A**) or means ± SEM (**B, C**) of technical replicates using the same lots of BPI and bacteria in three separate assays (**A, B**). Relative colony-forming units (CFUs) are depicted (**A–C**). Statistical testing was performed using the Student's paired *t*-test on the absolute CFUs (**A–C**). Statistical significance is indicated by p-values.

The online version of this article includes the following source data and figure supplement(s) for figure 4:

**Source data 1.** Raw data for **Figure 4A–C**.

*Figure 4 continued on next page*

*Figure 4 continued*

**Figure supplement 1.** Bactericidal activity of human bactericidal/permeability-increasing protein (huBPI) and scorpionfish BPI (scoBPI) against individual multiple drug resistance (MDR) isolates of *Pa* corresponding to data shown in *Figure 4*.

**Figure supplement 1—source data 1.** Raw data for *Figure 4—figure supplement 1A and B*.

The time-dependent bactericidal action of scoBPI against the MDR *Pa* isolates was also visualized in bactericidal viability assays (*Figure 4D*). Corresponding transmission electron microscopy (TEM) of a representative scoBPI- and huBPI-treated MDR *Pa* isolate revealed severe damage to the bacterial cell wall and efflux of cytoplasm in affected bacteria (*Figure 4E*, *Figure 4—figure supplement 1C and D*). In conclusion, scoBPI is bactericidal towards MDR *Pa* isolates derived from chronically infected PwCF.

## Discussion

Here, we demonstrate that scoBPI, an orthologous protein to huBPI, escapes recognition by BPI-ANCA of PwCF, displays high anti-inflammatory potency towards LPS, and exhibits bactericidal activity against *Pa*, including MDR *Pa* derived from PwCF. Indeed, we even observed a trend towards superior antibiotic activity against MDR *Pa* for scoBPI compared to huBPI at a concentration of 500 nM. As opposed to established antibiotics that mainly target processes associated with bacterial cell envelope biogenesis, DNA replication, transcription, and protein biosynthesis, the mode of action of BPI relies on membrane disruption by binding to LPS of the outer cell wall in Gram-negative bacteria (*Wiese et al., 1997*). Bactericidal activity via membrane perturbation of MDR *Pa* by scoBPI could be verified by TEM and explains the susceptibility to BPI despite MDR. As seen by viability imaging, the bactericidal activity of scoBPI was impressively fast and occurred within 1–3 hr. Activity via membrane permeabilization implies inhibition of non-replicating bacteria, possibly also including persister mutants of *Pa* commonly found in CF patients (*Bartell et al., 2020*). Sustained immune responses towards *Pa* in PwCF trigger a mucoid phenotype that is associated with resistance to host antimicrobials (*Malhotra et al., 2018*), impaired phagocytosis (*Cabral et al., 1987*), and formation of biofilms (*Høiby et al., 2010*), allowing for bacterial persistence in the CF environment. As shown previously for huBPI (*Aichele et al., 2006*), mucoid *Pa* isolates were indeed highly susceptible to the bactericidal action of scoBPI. Beyond CF, ESKAPE pathogens (*Enterococcus faecium*, *Staphylococcus aureus*, *Klebsiella pneumoniae*, *Acinetobacter baumannii*, *Pa*, and *Enterobacter* spp.) were defined by the WHO as priority pathogens for new antimicrobial drug development as these are continuously associated with multiple drug resistance and severe nosocomial infections (*Rice, 2008*). In this aspect, further studies of scoBPI on the Gram-negative members of the ESKAPE cluster, such as *Pa*, *Acinetobacter baumannii*-complex, and various Enterobacteriaceae, are worthwhile. It is noteworthy that BPI-ANCA are also frequently found in other inflammatory disorders like non-CF bronchiectasis (*Theprungsirikul et al., 2021b*), vasculitis (*Zhao et al., 2008*), and systemic lupus erythematosus (SLE; *Manolova et al., 2001*). Since pulmonary *Pa* infections are common in non-CF bronchiectasis

**Table 1.** Characteristics of PwCF and MDR *Pa* isolates.

| Isolate | Age at sampling (years) | Sex | Comment | Ciprofloxacin | Piperacillin ± Tazobactam | Ceftazidime/Cefepime | Imipenem/ Meropenem |
|---------|-------------------------|-----|---------|---------------|---------------------------|----------------------|---------------------|
| 1 | 31.7 | M | Mucoid | R | R | R | R |
| 2 | 22.9 | F | -/- | R | R | R | R |
| 3 | 31.0 | F | -/- | R | R | R | R |
| 4 | 12.1 | M | Mucoid | R | R | R | R |
| 5 | 26.2 | F | -/- | R | R | R | R |
| 6 | 24.0 | F | -/- | R | R | R | R |

Resistance was determined according to zone diameter breakpoints for *Pa* (EUCAST, version 13.0, published December 2022). Isolates 2 and 5 were derived from one individual patient with a time lag of 3.3 y.

MDR, multiple drug resistance; PwCF, people with cystic fibrosis; R, resistance to indicated antibiotic.

(*Theprungsirikul et al., 2021b*) and Gram-negative infections are known to be a leading cause of hospitalization in SLE patients (*Teh et al., 2018*), scoBPI might be of interest in patient groups other than CF as well. Administration of the recombinant human rBPI23 (amino acids 1–199) and rBPI21 (amino acids 1–193) has previously been tested in clinical phase I–III studies (*Levin et al., 2000*; *von der Mohlen et al., 1995*). Thereby, rBPI23 efficiently neutralized the endotoxic activity of LPS in healthy volunteers (*von der Mohlen et al., 1995*). Despite the peracute disease course of pediatric meningococcal sepsis and a limited number of patients, rBPI21 was shown to improve clinical outcome by trend (*Levin et al., 2000*). In a less acute setting, such as chronic persisting or exacerbation of *Pa* infections, the use of rBPI21, scoBPI, or the N-terminal barrel of scoBPI might even be more promising. Despite long-term infections in the tested MDR *Pa* and varying sensitivities, complete resistance to BPI was not found. Clearly, further studies should address this issue in a larger cohort. As suggested by our PBMC data, scoBPI potently neutralized the inflammatory potential of *Pa*-derived LPS and consequently prevented LPS-induced release of TNF and IL-6. Chronic pulmonary inflammation arises early in PwCF, leading to tissue destruction, and eventually accounts to respiratory failure and the premature deaths of patients (*Cantin et al., 2015*). The abundance of neutrophil-derived elastase in the lungs of PwCF is considered as the essential mediator of this persistent local inflammation (*Kelly et al., 2008*) and there is evidence that elastase release is primed by TNF (*Taggart et al., 2000*). It is also known that macrophages in PwCF are hyperresponsive to LPS, further driving dysregulated inflammatory responses (*Zhang et al., 2013*). Therefore, the benefits of administration of scoBPI by joining antimicrobial activity and LPS neutralization, including reduction of TNF release, could potentially extenuate inflammation and, thereby, tissue destruction. In future, BPI-ANCA could be purified and concentrated from sera or bronchoalveolar fluid of PwCF to show that these autoantibodies indeed selectively inhibit the bactericidal and LPS-neutralizing activity of huBPI but not scoBPI. Moreover, efficiency of scoBPI should be studied in vivo, for example, in CF and non-CF mouse models. Another interesting issue for further studies is whether LPS modifications commonly found in CF *Pa* isolates alter the interaction with BPI. These modifications include changes of the lipid A moiety by addition of 4-amino-4-deoxy-L-arabinose or changes in its fatty acid composition (*Ernst et al., 1999*), loss of the O-antigen and changes to the core oligosaccharide (*Knirel et al., 2001*). While LPS binding and the antimicrobial activity of BPI are mediated by the N-terminal barrel of the protein (*Little et al., 1994*), the C-terminal barrel is known to enhance phagocytosis by opsonization (*Iovine et al., 1997*). Fittingly, murine *Bpi*-deficient neutrophils display decreased phagocytosis of *Pa* in pulmonary infection models which could be reconstituted by exogenous BPI (*Theprungsirikul et al., 2021a*). Future in vitro studies will elucidate whether the C-terminal barrel of scoBPI also opsonizes *Pa* for phagocytosis by human cells. Concerning clinical use, one limitation might be a possible induction of antibodies against scoBPI during long-term or repeated applications. Therefore, therapeutic use may rather be indicated in short-term settings. Moreover, the starlet *A. ruthenus* (Actinopterygii, subclass Chondrostei) is only distantly related to the scorpionfish *S. schlegelii* (Actinopterygii, subclass Neopterygii). Congruently, acBPI shows only low sequence identity to scoBPI, thus low epitope overlap. Surprisingly, we found a similar positively charged cluster at the N-terminal tip of both orthologues. Therefore, Actinopterygii BPI other than scoBPI may also bear therapeutic potentials and should be tested functionally.

Collectively, we define scoBPI as a distantly related, orthologous protein to huBPI that combines LPS-neutralizing capacity and bactericidal activity towards MDR *Pa* isolates. Further investigations on scoBPI and other BPI orthologues are highly encouraged to develop novel therapeutics against Gram-negative MDR infections.

## Methods
### Expression and purification of recombinant BPI orthologues
Templates for recombinant BPI were designed by flanking the corresponding BPI sequences without native signal peptides (huBPI: amino acids 32–487; muBPI: amino acids 28–483; scoBPI: amino acids 19–473; osBPI: amino acids 20–477) with an N-terminal HA-signal peptide and a C-terminal FLAG-tag. Constructs were ligated into a pcDNA 3 (huBPI, muBPI) or pcDNA 3.1(+) (osBPI, scoBPI; Cat# V79020; Thermo Fisher Scientific, Waltham, MA) vector backbone. BPI orthologues were generated as described previously with slight modifications (*Ederer et al., 2022*; *Bülow et al., 2018*). Expi293F

cells (Cat# A14527, RRID:CVCL_D615; Thermo Fisher Scientific) were transfected using the ExpiFect-amine 293 Transfection Kit (Thermo Fisher Scientific). Recombinant huBPI, muBPI, and scoBPI were purified by cation-exchange chromatography on a HiTrap SP HP column (Cytiva, Marlborough, MA) followed by size-exclusion chromatography (Superdex 200 increase 10/300 GL column; Cytiva). Due to insufficient binding to the cation-exchange column, osBPI was purified by affinity chromatography on an anti-FLAG (Sigma-Aldrich, Taufkirchen, Germany) coupled NHS-activated HP column (Cytiva) followed by size-exclusion chromatography.

## Western blotting

For western blotting, 12 µl of purified recombinant orthologous proteins at a concentration of 250 nM were loaded onto a TGX FastCast 12% stain-free gel (Bio-Rad, Feldkirchen, Germany). Electrophoresis was conducted at 220 V for 30 min. Proteins were blotted onto a nitrocellulose membrane using the Trans-Blot Turbo TRA Transfer Kit (Bio-Rad). Following protein transfer, membranes were blocked in 5% non-fat dried milk dissolved in Tris-buffered saline with 0.05% Tween20 (TBS-T) for 1 hr. Anti-FLAG M2 antibody (Cat# F3165, RRID:AB_259529; Sigma-Aldrich) or polyclonal anti-human BPI antibody (Cat# HM2170, RRID:AB_532911; Hycult Biotech, Uden, Netherlands) were incubated overnight at 4°C. After three washes with TBS-T, the corresponding secondary, peroxidase-labeled antibodies rabbit anti-mouse (Cat# 711-035-152, RRID:AB_10015282; Dianova, Hamburg, Germany) and donkey anti-rabbit (Cat# 315-035-048, RRID:AB_2340069; Dianova) were incubated for 45 min at room temperature (RT) with three consequent washing steps. Membranes were then incubated with Clarity Western ECL substrate (Bio-Rad) for 5 min. Chemiluminescence was detected with the digital imaging system Chemi Lux Imager (Intas, Göttingen, Germany).

## Bacteria and antimicrobial resistance testing

*Ec* DH10B (Cat# EC0113; Thermo Fisher Scientific) and *Pa* PAO1 (ATCC15692, Cat# DSM 22644; DSMZ, Braunschweig, Germany) were cultivated on Columbia blood agar plates (Thermo Fisher Scientific). Antimicrobial resistance of MDR *Pa* strains was determined according to zone diameter breakpoints for *Pa* (EUCAST, version 13.0, published December 2022) in the diagnostic laboratory of the Institute of Clinical Microbiology and Hygiene Regensburg (University Hospital Regensburg, Germany). Microbial identification of all strains was confirmed by Matrix-Assisted Laser Desorption/Ionization Time-Of-Flight Mass Spectrometry (MALDI-TOF MS) using the MALDI Biotyper system (Bruker Corporation, Billerica, MA).

## Preparation of LPS

Isolation and purification of *Pa* PAO1 LPS was performed following previously described procedures (*Kutschera et al., 2019*). In detail, bacteria were grown aerobically with shaking (180 rpm) at 37°C in LB-Broth Base medium (Thermo Fisher Scientific) supplemented with 5 g/l of NaCl until an absorbance of approximately 1.1 at 600 nm was reached. Phenol (90%) was added to reach a final concentration of 1% and the resulting suspension was shaken (90 rpm) for 1 hr at 37°C and then at 4°C overnight. Cells were collected by centrifugation (9000 × *g*, 20 min, 4°C) and subsequently washed three times with water (centrifugation conditions as above). The lyophilized pellet (recovery, 2.78 g) was washed with ethanol, acetone (twice), and diethyl ether, and then dried. The pellet was resuspended in water (approximately 14 mg/ml), sequentially treated for 24 hr each at RT with DNase/RNase and proteinase K (100 µl of 10 mg/ml solutions per gram dry weight for each enzyme), then underwent dialysis (14 kDa cutoff) and lyophilization (yielding in 610 mg bacterial mass). For hot phenol-water extraction (*Westphal and Jann, 1965*), bacteria were resuspended in 45% aqueous phenol (40 ml per g bacteria) and stirred for 30 min at 68°C. After centrifugation (5600 × *g*) for 20 min at 4°C, the upper water phase was collected. The extraction was repeated with the same volume of water as had been collected. Combined water phases and the phenolic phase were dialyzed against water at RT (14 kDa cutoff). Prior to lyophilization, the dialyzed phenolic phase (PP) was centrifuged (600 × *g* for 5 min at 20°C) and divided into supernatant and sediment. The main part of the LPS was recovered from the water phase (184 mg). A portion (20.9 mg) of this LPS preparation was further purified by gel permeation chromatography on Sephacryl S-400 HR (Cytiva, Marlborough, MA, USA) on a column (2.5 × 120 cm) using a 50 mM ammonium bicarbonate buffer as eluent (*Jimenez-Barbero et al., 2002*), yielding 6.8 mg. Such material was used in the described experiments.

## Dynamic light scattering

BPI-LPS aggregate size was determined by DLS. Stock solutions of LPS derived from *Pa* PAO1 (500 μg/ml), huBPI and scoBPI (5 μM each) were pre-incubated alone or in the indicated combinations for 30 min at RT before filling the capillaries. Light scattering measurements were performed at 20°C using Prometheus Panta (NanoTemper Technologies GmbH, Munich, Germany).

## Stimulation of human PBMCs

Isolation of human PBMCs was performed as described previously (*Ederer et al., 2022*). In detail, blood from healthy volunteers was collected in heparinized tubes (Li-Heparin-Gel-Monovette, Sarstedt, Nümbrecht, Germany). The blood was centrifuged in Leucosep tubes containing Ficoll-Paque PLUS (Cytiva Europe GmbH, Freiburg, Germany) at $1000 \times g$ for 10 min. Leukocytes were collected from the interphase and subsequently washed twice with RPMI medium 1640 (Thermo Fisher Scientific). The PBMC pellet was then resuspended in AIM V (Thermo Fisher Scientific) and $10^5$ PBMCs were seeded into 96-well plates (Sarstedt) and rested for 3 hr prior to stimulation. Stimulants were diluted in AIM V, combined as indicated and pre-incubated in protein LoBind tubes (Eppendorf, Hamburg, Germany) for 30 min at 37°C. PBMC supernatants were collected after 24 hr and stored at –20°C for cytokine analysis.

## Quantification of cytokines, BPI, and BPI-ANCA

Levels of human IL-6 and TNF were quantified by Luminex Technology (Austin, TX) as described previously (*Ederer et al., 2022*). Capture and detection antibodies for human IL-6 were from the OptEIA set for human IL-6 (Cat# 55220, RRID:AB_2869045; BD Biosciences, Heidelberg, Germany), and capture and detection antibodies for human TNF were from the OptEIA set for human TNF (Cat# 555212, RRID:AB_2869042; BD Biosciences). Signals were detected by Streptavidin-PE (Agilent, Palo Alto, CA), and cytokine concentrations were quantified with the Human Cytokine 16-Plex Protein Standard (Thermo Fisher Scientific). For quantification of BPI in human sera, Luminex-beads coupled with anti-BPI antibody 3F9 (Hycult Biotech) were incubated with patient or control sera overnight at 4°C. BPI was detected with biotinylated anti-BPI antibody 4H5 (Cat# HM2042, RRID:AB_532909; Hycult Biotech) and Streptavidin-PE (Agilent). Concentrations were quantified with BPI as a standard extracted from neutrophils (Wieslab AB, Malmö, Sweden). For measurement of BPI-ANCA, recombinant BPI orthologues were coupled to beads and the respective coupling efficiency was controlled by incubation of beads with anti-FLAG M2 antibody (Cat# F3165, RRID:AB_259529; Sigma-Aldrich) for 60 min followed by incubation with anti-mouse-PE (#715-065-140, Dianova, Eching, Germany) for 30 min (*Figure 2—figure supplement 1D*). After the incubation with sera PwCF and healthy controls, binding of autoantibodies was detected by a polyclonal PE-labeled anti-human IgG, Fcγ fragment-specific antibody (Cat# 109-116-098, RRID:AB_2337678; Dianova). The signal cutoff for positivity was determined using sera of 39 healthy age- and sex-matched controls and defined as 2 standard deviations above the mean signal. Autoantibody levels are indicated as AU. Cytokine and autoantibody levels were measured using the Luminex 100 system (Austin) and subsequently analyzed by using LiquiChip Analyzer Software (QIAGEN, Hilden, Germany).

## Dose–response experiments

Dose–response experiments were conducted as described previously (*Ederer et al., 2022*). In brief, bacteria were grown in LB medium at 37°C and 220 rpm to an optical density ($OD_{600}$) of 0.4. Bacteria were further diluted in PBS (Sigma-Aldrich) with 0.01% Tween20 (PBS-T) and incubated with recombinant BPI or PBS-T as mock control for 1 hr at 37°C without shaking. Dilutions of bacteria were then plated onto blood agar plates and incubated at 37°C overnight. Colony-forming units were enumerated the next day. For MDR isolates of *Pa*, bacteria were incubated with 20 nM and 500 nM BPI, respectively.

## Bacterial viability assay

For bacterial viability assays, the LIVE/DEAD *Bac*Light Bacterial Viability Kit from Thermo Fisher Scientific was used. *Pa* isolates were grown to an $OD_{600}$ of 0.2–0.8, pelleted and resuspended in PBS to a final $OD_{600}$ of 1.0. Next, 50 μl of cell suspension was diluted with 10 μl of scoBPI to a final scoBPI

concentration of 500 nM. For excitation of dyes and microscopic imaging, the Keyence All-in-one Fluorescence Microscope BZ-9000 (Keyence, IL) was used.

## Transmission electron microscopy

Bacteria were grown to an $OD_{600}$ of 0.4. Bacteria were pelletized and resuspended in PBS-T to an $OD_{600}$ of 10. Then, $5 \times 10^9$ bacteria were incubated with 2 µM scoBPI for 2 hr. Bacteria were pelletized, the supernatant replaced by cacodylate-buffered 2.5% glutaraldehyde solution (all reagents from EMS, Science Services, Munich, Germany) and the samples were fixed for 12 hr at RT. Alginate (Epredia Cytoblock Replacement Reagents, Thermo Fisher Scientific) was used to enhance pellet stability for the untreated control. After that, bacteria treated with scoBPI and untreated control were enclosed with 4% low melting agarose. The Lynx microscopy tissue processor (Reichert-Jung, Wetzlar, Germany) was used for the embedding process, including a secondary fixation with 1% osmium tetroxide solution (EMS, Science Services), dehydration and infiltration with EPON-mixture (Embed, DDSA, NMA, DP30). The EPON-infiltrated pellets were transferred in silicon molds and overlaid with fluid EPON-mixture and harden at 60°C in the heating oven. Semithin sections (0,75 µm) were used to define relevant areas and ultrathin sections (80 nm) were then cut by use of the Reichert Ultracut S (Leica Microsystems, Wetzlar, Germany). The ultrathin sections were then contrasted with aqueous 2% uranyl-acetate (EMS, Science Services) and 2% lead-citrate solution (Leica Ultrostain II, Leica Microsystems) for 10 min each. Electron microscopic analysis was performed using the LEO 912 AB electron microscope (Zeiss, Oberkochen, Germany).

## Sequence analysis, alignment, and homology calculation

Analyzed protein sequences were from UniProt (*The UniProt Consortium, 2021*; huBPI: P17213, muBPI: Q67E05, scoBPI: A0A481NSZ4, osBPI: C4NY84). Sequences were aligned using Clustal Omega (*Sievers et al., 2011*). Sequence identities were calculated with SIM-alignment tool for protein sequences (*Huang and Miller, 1991*).

## Structure modeling, graphical depictions, and statistical analysis

The Protein Data Bank (PDB) structure for huBPI (PDB DOI:10.2210/pdb1BP1/pdb ) deposited by *Beamer et al., 1997* and AlphaFold Protein Structure Database (*Jumper et al., 2021*; *Varadi et al., 2022*) predictions for muBPI, scoBPI, and osBPI were used for three-dimensional modeling. Three-dimensional structures were rendered in PyMOL (PyMOL Molecular Graphics System, version 2.3.2 Schrödinger, LLC, New York, NY), and electrostatics were calculated using the APBS plugin for PyMOL. Depictions of graphs and statistical analyses were done using GraphPad Prism, version 7.01 (GraphPad Software, San Diego, CA). Results are depicted as indicated in the figure legends as absolute values, means ± SEM or box plots showing median, upper, and lower quartiles and whiskers indicating minimal and maximal values. Statistical tests were performed as described in the figure legends. Thereby *t*-test was chosen when values were normally distributed. p-values<0.05 were considered statistically significant. Values <0 of the lower 95% CI were set to 0.

## Acknowledgements

We gratefully acknowledge Sabine Markowski and Lisa Zeller (both Institute of Clinical Microbiology and Hygiene, Regensburg, University Hospital Regensburg, Germany), Claudia Fischer (Institute of Pathology, University of Regensburg, Germany), as well as Michelle Wröbel and Ursula Schombel (both Research Center Borstel, Germany) for technical support. We thank Dr. Uwe Mamat (Research Center Borstel, Germany) for providing the *Pa* PAO1 strain for preparation of LPS.

## Additional information

### Competing interests

André Gessner, Sigrid Bülow: The University of Regensburg is applying for a patent (PCT/EP2022/087280) covering parts published in this manuscript with André Gessner and Sigrid Bülow as inventors. The other authors declare that no competing interests exist.

## Funding

No external funding was received for this work.

## Author contributions

Jonas Maurice Holzinger, Data curation, Formal analysis, Methodology, Visualization, Writing – original draft; Martina Toelge, Data curation, Formal analysis, Methodology, Visualization; Maren Werner, Heiko Ingo Siegmund, David Peterhoff, Data curation; Katharina Ursula Ederer, Visualization; Stefan Helmut Blaas, Methodology; Nicolas Gisch, Methodology, Resources; Christoph Brochhausen, Resources; André Gessner, Data curation, Methodology, Writing – review and editing; Sigrid Bülow, Conceptualization, Data curation, Formal analysis, Investigation, Supervision, Validation, Writing – review and editing

## Author ORCIDs

Jonas Maurice Holzinger http://orcid.org/0009-0007-3274-7837
Nicolas Gisch http://orcid.org/0000-0003-3260-5269
André Gessner http://orcid.org/0000-0003-4316-2408
Sigrid Bülow http://orcid.org/0000-0002-8952-290X

## Ethics

This study was carried out in accordance with the recommendations of the Declaration of Helsinki. Diagnostic leftover samples, stored at the Institute of Clinical Microbiology and Hygiene, University Hospital Regensburg, were used for serum analysis. PBMCs were purified from blood of healthy donors. All donors of PBMCs gave written informed consent. The protocols were approved by the local ethics committee (Ethikkommision an der Universität Regensburg, 18-1269-101 and 16-302_1-101).

## Decision letter and Author response

Decision letter https://doi.org/10.7554/eLife.86369.sa1
Author response https://doi.org/10.7554/eLife.86369.sa2

---

# Additional files

## Supplementary files

• MDAR checklist

## Data availability

All data generated or analysed during this study are included in the manuscript and supporting file. Source data files have been provided for Figures 2, 3 and 4.

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
