## [Editor Report]

In this useful study, Holzinger et al. present compelling evidence that scorpionfish bactericidal/permeability-increasing protein (scoBPI) exhibits remarkable antibacterial activity against multi-drug resistant *Pseudomonas aeruginosa*. These findings open new avenues of research for identifying novel chemotherapies to treat *Pseudomonas* infections and have broader implications in developing chemotherapies against other drug-resistant Gram-negative bacterial infections. The work will be of interest to individuals investigating novel cystic fibrosis antimicrobials.

---

## [Decision Letter]

**Decision letter after peer review:**

Thank you for submitting your article "Scorpionfish BPI is highly active against multiple drug-resistant *Pseudomonas aeruginosa* isolates from cystic fibrosis patients" for consideration by *eLife*. Your article has been reviewed by 2 peer reviewers, and the evaluation has been overseen by a Reviewing Editor and Bavesh Kana as the Senior Editor. The following individuals involved in the review of your submission have agreed to reveal their identity: Thomas Barton (Reviewer #1); Avishek Mitra (Reviewer #2).

Essential revisions:

1. Throughout the report, it would be more appropriate to refer to CF patients as people with CF (PwCF).

2. Figure 2B, C – Who are the healthy controls? Are they age/sex-matched to the people with CF?

3. Figure 2 – An experiment with the same conditions in the presence of anti-BPI antibodies would more effectively demonstrate the impact of scoBPI, as one would expect little cytokine reduction with huBPI compared to scoBPI, which would be expected to evade sequestration by anti-BPI.

4. Figure 2E – What controls were used?

5. Figure 4E – What is the timepoint?

6. Figure 3/4 – Cytokine expression and CFU are displayed as percentages. The study would benefit from displaying raw values, as differences in percentages do not necessarily translate to meaningful experimental outcomes.

7. Lines 72-74, beginning "Upon 10", are confusingly worded.

8. How was the threshold (dashed line) determined in Figure 2E? Is it the mean of the huBPI group? Are the data for huBPI identical to those for the CF group in Figure 2C or was the assay in 2C performed differently?

9. CF Pa isolates often have modified LPS (e.g. by the addition of L-Ara-4N to lipid A). This can alter the LPS charge, potentially diminishing interaction with BPI. Did the authors test the LPS-neutralisation activity of scoBPI with LPS purified from CF isolates?

10. In Figure 4C it would be desirable to be able to identify the isolate associated with each individual line. The current styling makes it challenging to differentiate strains based on the given description.

11. Line 216 – The claim of activity against persisters is not supported by any data. Either this should be tested or the sentence should be removed.

12. Line 218 – Should adaption be adoption?

13. Provide gel images and chromatograms of purified recombinant proteins.

14. Using the in cidal assay as shown in Figure 4, determine:

– Whether adding BPI-ANCA inhibits cidal activity of huBPI and not scoBPI.

– Based on the binding results shown in Figure 2F, BPI-ANCA should inhibit huBPI cidal activity. This would provide an important control and further support the hypothesis.

15. For the dose-response assays, describe in methods if cells were incubated with shaking or statically during BPI incubation.

16. Include huBPI as a control for the TEM experiments.

*Reviewer #1 (Recommendations for the authors):*

– Throughout the report, it would be more appropriate to refer to CF patients as people with CF (PwCF).

– Figure 2B, C – Who are the healthy controls? Are they age/sex-matched to the people with CF?

– Figure 2 – An experiment with the same conditions in the presence of anti-BPI antibodies would more effectively demonstrate the impact of scoBPI, as one would expect little cytokine reduction with huBPI compared to scoBPI, which would be expected to evade sequestration by anti-BPI.

– Figure 2E – What controls were used?

– Figure 4E – What is the timepoint?

– Figure 3/4 – Cytokine expression and CFU are displayed as percentages. The study would benefit from displaying raw values, as differences in percentages do not necessarily translate to meaningful experimental outcomes.

– Lines 72-74, beginning "Upon 10", are confusingly worded.

– How was the threshold (dashed line) determined in Figure 2E? Is it the mean of the huBPI group? Are the data for huBPI identical to those for the CF group in Figure 2C or was the assay in 2C performed differently?

– CF Pa isolates often have modified LPS (e.g. by addition of L-Ara-4N to lipid A). This can alter the LPS charge, potentially diminishing interaction with BPI. Did the authors test the LPS-neutralisation activity of scoBPI with LPS purified from CF isolates?

– In Figure 4C it would be desirable to be able to identify the isolate associated with each individual line. The current styling makes it challenging to differentiate strains based on the given description.

– Line 216 – The claim of activity against persisters is not supported by any data. Either this should be tested or the sentence should be removed.

– Line 218 – Should adaption be adoption?

*Reviewer #2 (Recommendations for the authors):*

Please consider the following points to strengthen the manuscript.

1. Provide gel images and chromatograms of purified recombinant proteins.

2. Please determine in cidal assay as shown in Figure 4:

– Whether adding BPI-ANCA inhibits cidal activity of huBPI and not scoBPI.

– Based on the binding results shown in Figure 2F, BPI-ANCA should inhibit huBPI cidal activity.

– This would provide an important control and further support your hypothesis.

3. For the dose-response assays, please describe in methods if cells were incubated with shaking or statically during BPI incubation.

4. Please consider including huBPI as a control for the TEM experiments.

---

## [Author Response]

Essential revisions:1. Throughout the report, it would be more appropriate to refer to CF patients as people with CF (PwCF).

That is right. The term “CF patients” was exchanged to “people with CF (PwCF)” throughout the whole manuscript.

2. Figure 2B, C – Who are the healthy controls? Are they age/sex-matched to the people with CF?

Sera of healthy controls were derived from anonymous blood donors. Thus, no data on sex or age was available. We therefore collected a new cohort of healthy controls which are now age- and sex-matched regarding the CF group. PwCF were pseudonymized and had to be de-pseudonymized for determination of age and sex. This made us aware that 7 sera originated from identical patients. Only the first serum was used in these cases (leaving 39 instead of 46 patients). We are sorry for the mistake.

All samples were (re-)measured in parallel, including samples from PwCF as well as from healthy controls. New data are added in the text and as source data. No difference in serum BPI levels was found between the groups in the age- and sex-matched data set. Yet, p-value for BPI-ANCAs between PwCF and controls remained at p°<°0.0001. Percent of positivity for BPI-ANCAs changed from 54.3% to 51.0%. Cross reactivity towards orthologues was unchanged (muBPI, scoBPI) or was even lower than previously found (2.6% instead of 6.5%, osBPI). In accordance to the new data, respective passages were corrected in the abstract, Figure 2 B, C and E, the legend to Figure 2 as well as the respective text passage.

“First, we screened the sera of 39 PwCF for presence of BPI and BPI-ANCA. While levels of BPI in sera were comparable among PwCF and an age- and sex-matched control group (Figure 2B), BPI-ANCA as measured in arbitrary units (AU) were significantly increased in PwCF (control: mean 314.8 AU, 95% confidence interval (CI) 209.3 – 420.4 AU; CF: mean 2218.0 AU, 95% CI 1271.0 – 3166.0 AU, p < 0.0001; Figure 2C).“

“While huBPI was bound by anti-BPI antibodies in 51.0%, muBPI and scoBPI were not recognized at all and osBPI was only detected by 2.6% of the CF sera (Figure 2E).”

3. Figure 2 – An experiment with the same conditions in the presence of anti-BPI antibodies would more effectively demonstrate the impact of scoBPI, as one would expect little cytokine reduction with huBPI compared to scoBPI, which would be expected to evade sequestration by anti-BPI.

This is an important and interesting point. As shown by McQuillan et al.^1^, BPI-ANCA purified from sera of PwCF can inhibit the bactericidal function of huBPI. To perform the requested experiment with BPI-ANCA derived from PwCF, an ethic statement would be necessary and methods for purification have to be established. Therefore, this experiment cannot be done within 2 months as required by the *eLife* guidelines.

As an alternative we tried to use a commercial rabbit polyclonal antibody obtained after vaccination with holo-BPI (Cat# HM2170, RRID: AB_532911; Hycult Biotech, Uden, Netherlands). Unfortunately, increase of IL-6 and TNF secretion was seen with the antibody alone and in combination with LPS derived from *Pseudomonas aeruginosa* PAO1 (Author response image 1). Both data sets indicate contamination of the antibody with proinflammatory substances. Although diluted, the sodium acid contained in the antibody (0.02%) might also influence the experiment. Therefore, a sodium acid free antibody tested for low endotoxin content would be necessary. To the best of our knowledge such an antibody is not commercially available. Additionally, a rabbit antibody might not reflect the effects of BPI-ANCA derived from PwCF. To our opinion, the use of currently commercially available non-human antibodies does not meet the standards required by *eLife*.

**Author response image 1. sa2fig1:** Background activity of a rabbit polyclonal anti-huBPI in PBMCs. (A) Levels of IL-6 and TNF in supernatants of PBMCs stimulated for 24 h with a polyclonal anti-huBPI antibody. (B) Levels of IL-6 and TNF in supernatants of PBMCs stimulated for 24 h with PAO1 LPS (10 ng/ml) or medium ± polyclonal anti-huBPI antibody. Experiments were performed using PBMCs of two (A) or five (B) individual blood donors. Data are shown as means ± SEM. Statistical testing was performed using the student’s ratio paired t test (A). Statistical significance is indicated by p values. NT: not treated.

We added an outlook in the discussion:

“In future, BPI-ANCA could be purified and concentrated from sera or bronchoalveolar fluid of PwCF to show that these autoantibodies indeed selectively inhibit huBPI but not scoBPI in concern of the bactericidal and LPS-neutralizing activity. Thereby, also in vivo studies, e.g., in CF and non-CF mouse models, should be included.”

4. Figure 2E – What controls were used?

Coupling controls for the BPI orthologues are now shown (Figure 2—figure supplement 1D).

5. Figure 4E – What is the timepoint?

The timepoint of two hours is stated in the methods section and was additional added in the legend to Figure 4.

6. Figure 3/4 – Cytokine expression and CFU are displayed as percentages. The study would benefit from displaying raw values, as differences in percentages do not necessarily translate to meaningful experimental outcomes.

Since we agree with that point, we added the respective raw values to the Source data files of Figure 3. Depiction of the raw values of Figure 3B, D and E can be seen in Author response image 2. If you agree, we would prefer to show the relative values to make estimation of the relative reduction in cytokine secretion in dependence of BPI visible (respective right graph in Figure 3B, D and E). Absolute values for the cytokine secretion caused by LPS alone were already included and remain in Figure 3 to get an idea of the absolute values measured in the individual donors (respective left graph in Figure 3B, D and E).

**Author response image 2. sa2fig2:** Potent anti-inflammatory action of scoBPI in human immune cells corresponding to data shown in Figure 3. (A) Absolute levels of IL-6 in supernatants of PBMCs stimulated for 24 h with Ec B4:O111 LPS (10 ng/ml) ± huBPI (red) or scoBPI (blue) in concentration as indicated. (B, C) Absolute quantification of IL-6 (B) and TNF (C) levels in supernatants of PBMCs stimulated for 24 h with Pa PAO1 LPS (100 ng/ml) ± huBPI (red) or scoBPI (blue) in concentrations as indicated (n = 3). Experiments were performed using PBMCs of four (A) or three (B, C) individual blood donors. Data are shown as means ± SEM. Statistical testing was performed using the student’s ratio paired t test. Statistical significance is indicated by p values.

Concerning Figure 4, the absolute CFUs are now shown in Figure 4—figure supplement 1A and B, the respective absolute raw values of Figure 4 A-C were added to the Source data files of Figure 4 and are also shown in Author response image 3. We hope you agree that the overall effects are meaningful when looking at the absolute values and depiction of relative values makes effects more clearly as all CFU start at the same location on the Y-axis.

**Author response image 3. sa2fig3:** Bactericidal activity of orthologous proteins corresponding to data shown in Figure 4. (A) Dose-response curves *Ec* DH10B incubated with increasing concentrations of huBPI, muBPI, scoBPI or osBPI (n = 3). (B) Bactericidal activity of huBPI, muBPI, scoBPI and osBPI 500 nM against *Pa* PAO1 (n = 3). (C) Antibacterial activity of huBPI and scoBPI at concentrations of 20 and 500 nM against six MDR isolates of *Pa* obtained from five individual PwCF. Isolates (isolate 2 and 5 in Table 1) that originate from the same donor are shown as a square. Mucoid isolates are displayed as filled symbols connected by continuous lines. Data are shown as individual points (A, C) or means ± SEM (B) of absolute values. Statistical testing was performed using the student’s paired *t* test. Statistical significance is indicated by *p* values.

7. Lines 72-74, beginning "Upon 10", are confusingly worded.

We agree. The sentence is now changed.

“In comparison to 9 other Actinopterygii BPI sequences, scoBPI comprised the highest number of the cationic amino acids arginine, histidine, and lysine in the corresponding regions (Figure 1—figure supplement 1A-C).”

8. How was the threshold (dashed line) determined in Figure 2E? Is it the mean of the huBPI group?

The determination of the threshold is stated in the methods section and was additionally added in the legend to Figure 2.

“The signal cut off is indicated by a dotted line and was defined as two standard deviations above the mean signal determined in sera of healthy age- and sex-matched controls shown in (C).”

Are the data for huBPI identical to those for the CF group in Figure 2C or was the assay in 2C performed differently?

You are right, these are duplicate data which we show for comparison reasons. We now clarify this in the figure legend.

“Recognition of recombinantly expressed proteins by anti-BPI antibodies present in the sera of the 39 individual PwCF shown in (C).”

9. CF Pa isolates often have modified LPS (e.g. by the addition of L-Ara-4N to lipid A). This can alter the LPS charge, potentially diminishing interaction with BPI. Did the authors test the LPS-neutralisation activity of scoBPI with LPS purified from CF isolates?

Several changes in the LPS structure derived from *Pseudomonas aeruginosa* of PwCF were described and we are very interested in their effect on the activity of the tested BPI orthologues. Still, LPS of several *Pseudomonas aeruginosa* isolates would have to be extracted, purified and analyzed by mass spectrometry. Therefore, we would rather feel that this would be a new project and would take several months. Thus, for now, we can only add this very interesting point in the discussion.

“Another interesting issue for further studies is whether LPS modifications commonly found in CF Pa isolates alter the interaction with BPI. These modifications include changes of the lipid A moiety by addition of 4-amino-4-deoxy-L-arabinose or changes in its fatty acid composition (48), loss of the O-antigen and changes to the core oligosaccharide (49).”

10. In Figure 4C it would be desirable to be able to identify the isolate associated with each individual line. The current styling makes it challenging to differentiate strains based on the given description.

To make the individual responses visible, we added the respective graphs in Figure 4—figure supplement 1A, B and referred to in the main text.

11. Line 216 – The claim of activity against persisters is not supported by any data. Either this should be tested or the sentence should be removed.

It is correct that we did not show any data on persisters. However, clear effects in non-replicating bacteria treated with scoBPI and huBPI are visible in TEM. Thus, we modified the text in order to make it clear that we only speculate.

“Activity via membrane permeabilisation implies inhibition of non-replicating bacteria, possibly also including persister mutants of *Pa* commonly found in CF patients (34).”

12. Line 218 – Should adaption be adoption?

We modified the text.

“Sustained immune responses towards *Pa* in PwCF trigger a mucoid phenotype that is associated with resistance to host antimicrobials (35), impaired phagocytosis (36) and formation of biofilms (37), allowing for bacterial persistence in the CF environment.”

13. Provide gel images and chromatograms of purified recombinant proteins.

We added the missing data in Figure 2—figure supplement 1A, B, and C and referred to it in the text.

14. Using the in cidal assay as shown in Figure 4, determine:– Whether adding BPI-ANCA inhibits cidal activity of huBPI and not scoBPI.– Based on the binding results shown in Figure 2F, BPI-ANCA should inhibit huBPI cidal activity. This would provide an important control and further support the hypothesis.

To perform the requested experiment with BPI-ANCA derived from PwCF, an ethic statement would be necessary and methods for purification have to be established. Therefore, this experiment cannot be done within 2 months as required by the *eLife* guidelines. We would also like to refer to comment 2 and our respective answer. In case of bactericidal tests, the sodium acid in the preparation of the commercially available polyclonal antibody might interfere with the bactericidal activity of BPI.

We added an outlook in the discussion:

“In future, BPI-ANCA could be purified and concentrated from sera or bronchoalveolar fluid of PwCF to show that these autoantibodies indeed selectively inhibit huBPI but not scoBPI in concern of the bactericidal and LPS-neutralizing activity. Thereby, also in vivo studies, e.g., in CF and non-CF mouse models, should be included.”

15. For the dose-response assays, describe in methods if cells were incubated with shaking or statically during BPI incubation.

Experiments were performed without shaking. This is clarified in the text now.

16. Include huBPI as a control for the TEM experiments.

A respective TEM image of huBPI on the *Pa* isolate shown in Figure 4 is now depicted in Figure 4—figure supplement 1D and referred to in in the text.